# Bioacoustics for *in situ* validation of species distribution modelling: An example with bats in Brazil

Frederico Hintze[1,2]*, Ricardo B. Machado[3], Enrico Bernard[1]

**1** Laboratório de Ciência Aplicada à Conservação da Biodiversidade, Departamento de Zoologia, Universidade Federal de Pernambuco, Recife, Pernambuco, Brasil, **2** Programa de Pós-graduação em Biologia Animal, Departamento de Zoologia, Universidade Federal de Pernambuco, Recife, Pernambuco, Brasil, **3** Departamento de Zoologia, Instituto de Ciências Biológicas, Campus Darcy Ribeiro, Universidade de Brasília, Brasília, Distrito Federal, Brasil

* fredhintze@gmail.com

**Data Availability Statement:** All relevant data are within the manuscript and its Supporting information files.

## Abstract

Species distribution modelling (SDM) gained importance on biodiversity distribution and conservation studies worldwide, including prioritizing areas for public policies and international treaties. Useful for large-scale approaches and species distribution estimates, it is a plus considering that a minor fraction of the planet is adequately sampled. However, minimizing errors is challenging, but essential, considering the uses and consequences of such models. *In situ* validation of the SDM outputs should be a key-step—in some cases, urgent. Bioacoustics can be used to validate and refine those outputs, especially if the focal species' vocalizations are conspicuous and species-specific. This is the case of echolocating bats. Here, we used extensive acoustic monitoring (>120 validation points over an area of >758,000 km$^2$, and producing >300,000 sound files) to validate MaxEnt outputs for six neotropical bat species in a poorly-sampled region of Brazil. Based on *in situ* validation, we evaluated four threshold-dependent theoretical evaluation metrics' ability in predicting models' performance. We also assessed the performance of three widely used thresholds to convert continuous SDMs into presence/absence maps. We demonstrated that MaxEnt produces very different outputs, requiring a careful choice on thresholds and modeling parameters. Although all theoretical evaluation metrics studied were positively correlated with accuracy, we empirically demonstrated that metrics based on specificity-sensitivity and sensitivity-precision are better for testing models, considering that most SDMs are based on unbalanced data. Without independent field validation, we found that using an arbitrary threshold for modelling can be a precarious approach with many possible outcomes, even after getting good evaluation scores. Bioacoustics proved to be important for validating SDMs for the six bat species analyzed, allowing a better refinement of SDMs in large and under-sampled regions, with relatively low sampling effort. Regardless of the species assessing method used, our research highlighted the vital necessity of *in situ* validation for SDMs.

**Funding:** FH was supported by a PhD grant from the Coordenação de Aperfeiçoamento de Pessoal de Nível Superior–Brasil (CAPES - https://www.gov.br/capes/pt-br) – Finance Code 001 EB and RBM have produtivity fellowships conceded by the Conselho Nacional de Desenvolvimento Científico e Tecnológico (CNPq - https://www.gov.br/cnpq/pt-br). The funders had no role in study design, data collection and analysis, decision to publish, or preparation of the manuscript.

**Competing interests:** The authors have declared that no competing interests exist.

## Introduction

Species distribution modelling (SDM) gained importance worldwide in the development of studies on biodiversity distribution and conservation [1, 2]. These distribution models can be produced and refined by crossing species presence records with biological and non-biological variables and environmental data [e.g., 3]. This is an advantage for large-scale approaches and species distribution estimates since few parts of our planet have been adequately sampled [4, 5]. Moreover, SDM has been used to support decision-making processes, including prioritizing areas and regions in public policies and international treaties [2, 5].

For biodiversity conservation purposes, imprecise models can undermine the calculation/estimate of a species' occupancy, a criterion used to assess its conservation status, for example [6]. Depending on the quality of the input data and modelling parameters chosen, the predictions created may not forecast precisely the species' distribution [7, 8]. Known as commission (false positives) and omission (false negatives) errors, they can inflate or reduce the potential distribution of a given *taxon*. Minimizing such errors is a challenge for spatial modelling [4, 7, 9], but essential considering the uses and consequences such models may have [10]. Therefore, *in situ* validation of the SDM outputs should be a critical step—in some cases, urgent [10, 11] since unvalidated species potential distribution maps can influence and hinder species assessments and the decision-making for species conservation.

Certifying that a species is present in a given area is not always straightforward, and different and innovative approaches have been proposed for such tasks [e.g., e-DNA [12] or satellite images [13]]. Bioacoustics is one of such techniques and has been used for a long time to record species presence/absence for amphibians, birds, and cetaceans [14]. Bats are a widespread group, using many habitats and resources, and most of the 1400 known species depend on echolocation for navigation and food acquisition [15]. The bats' echolocation system is based on ultrasonic signals which, although above the human hearing capacity, can be easily recorded thanks to recent electronics advances [16]. Moreover, with the exception of the Phyllostomidae family, most bats have conspicuous and species-specific calls, allowing precise identification of the emitter's identity and making the record of its presence accurate [16, 17]. Therefore, bioacoustics can be a useful technique applied to the *in situ* validation and refinement of SDM outputs for echolocating bats.

Here, we used an extensive *in situ* monitoring (>120 validation points over an area of >758,000 km2, and producing >300,000 sound files) of echolocation calls to validate the outputs of the most used SDM algorithm [MaxEnt; 3] for six neotropical bat species in a poorly-sampled part of Brazil; aiming to the evaluate SDM's performance of using different thresholds given the validation dataset obtained with bioacoustics. Using independent acoustic data collected, we (a) evaluated the ability of four threshold-dependent theoretical evaluation metrics in predicting models' performance, and (b) assessed the ability of three widely used thresholds to convert continuous species habitat suitability models into binary (presence/absence) maps. This methodological procedure enabled us to assess of the role of validation methods in SDM outputs and acoustic samplings as rapid validation method.

## Materials and methods

### Historical species records

We selected six neotropical bat species whose echolocation calls are well-known, species-specific and unequivocally identifiable: *Noctilio leporinus*, *Promops centralis*, *Promops nasutus*, *Pteronotus gymnonotus*, *Pteronotus personatus*, and *Saccopteryx leptura* [18–20]. We gathered distribution records for these species from a bibliographic revision using the following online

databases and search engines: the Vertebrate Zoology Database of the American Museum of Natural History (https://sci-web-002.amnh.org/db/emuwebamnh/index.php); the database of the Division of Mammals Collections of the Smithsonian National Museum of Natural History (https://collections.nmnh.si.edu/search/mammals/); the Global Biodiversity Information Facility (www.gbif.org); the SpeciesLink network (http://www.splink.org.br); Google Scholar (scholar.google.com); the Web of Science (www.webofknowledge.com); Scopus (www.scopus.com); the Periódicos CAPES (www.periodicos.capes.gov.br); and the Scientific Electronic Library Online (www.scielo.br). For that review, we searched for publications using keywords in the search engines such as: "neotropical bats", "Pteronotus", "Noctilio", "Promops", "Saccopteryx", "Pteronotus personatus", "P. personatus" and so on. We did not refined the searches by any specific field areas nor geographical areas. We also reviewed occurrences in studies available online such as Barquez, Ojeda [21] and Gardner [22]. We only selected bat records from peer-reviewed literature, books, or online databases supported by voucher specimens. Each record was checked for duplication of localities and, eventually, to correct location or taxonomy problems [23]. For example, following Gardner [22], we treated *P. nasutus* as a monotypic species (considering *Promops nasutus ancilla*, *P. nasutus pamana*, *P. nasutus fosteri*, or *P. nasutus downsi* as simply *Promops nasutus*). Overall, we gathered a total of 1277 single records for the six studied bat species: 590 records for *Noctilio leporinus*, 70 for *Promops centralis*, 71 for *Promops nasutus*, 120 for *Pteronotus gymnonotus*, 95 for *Pteronotus personatus*, and 331 for *Saccopteryx leptura* (Fig 1, and S1 Table).

## Distribution modelling procedure

We used SDMtoolbox 2.4 for ArcGIS [24] to create an environmental heterogeneity map with all bioclimatic variables from WorldClim 2.0 [25]. To reduce the potential bias caused by autocorrelation, we then used the Spatial Rarefy Occurrence Data tool of the SDMtoolbox 2.4 package to delete records under the same environmental conditions within 25 km from each other [26]. Other studies usually use a 10km distance between points [e.g. 27–29], but given the topographic and environmental heterogeneity of the studied region, we chose 25 km as a spatial filter to avoid eventual spatial bias on historical records [30]. This process ensured that all historical records gathered in our revision corresponded to a unique spatial sample, reducing occurrence data from 1277 to 899 single localities: 375 localities for *Noctilio leporinus*, 61 for *Promops centralis*, 65 for *Promops nasutus*, 87 for *Pteronotus gymnonotus*, 66 for *Pteronotus personatus*, and 245 for *Saccopteryx leptura*.

We used MaxEnt 3.4 [3] to generate potential species distribution models for the six selected species based on a set of variables at a 5 km$^2$ resolution. Due to the shortfalls and constraints of absences and pseudo-absences in the species distribution knowledge, and consequently, in species distribution models, we chosen MaxEnt as it uses presence-only input data, can include both categorical and continuous covariables, and create a spatially explicit suitability map for the focal species. We used the 19 bioclimatic variables plus elevation available at the Worldclim data website [25], and Globcover 2009 [31] as a categorical variable for land cover. To reduce the multicollinearity among the predictor variables, we performed a preliminary model with all variables and checked the weight of each according to their contributions, using Jackknife tests. Next, using the Correlations and Summary Stats tool of the SDMtoolbox 2.4 package, we obtained the correlation and covariances matrices and removed highly correlated variables (i.e., those with the lowest value when the pairwise correlation was > 0.7) [32]. Therefore, we used different variables for each modelled species (S2 Table).

We used a logistic output to produce our models and obtain continuous suitability values for species habitat suitability, which varies from 0 (lowest suitability) to 1 (highest suitability)

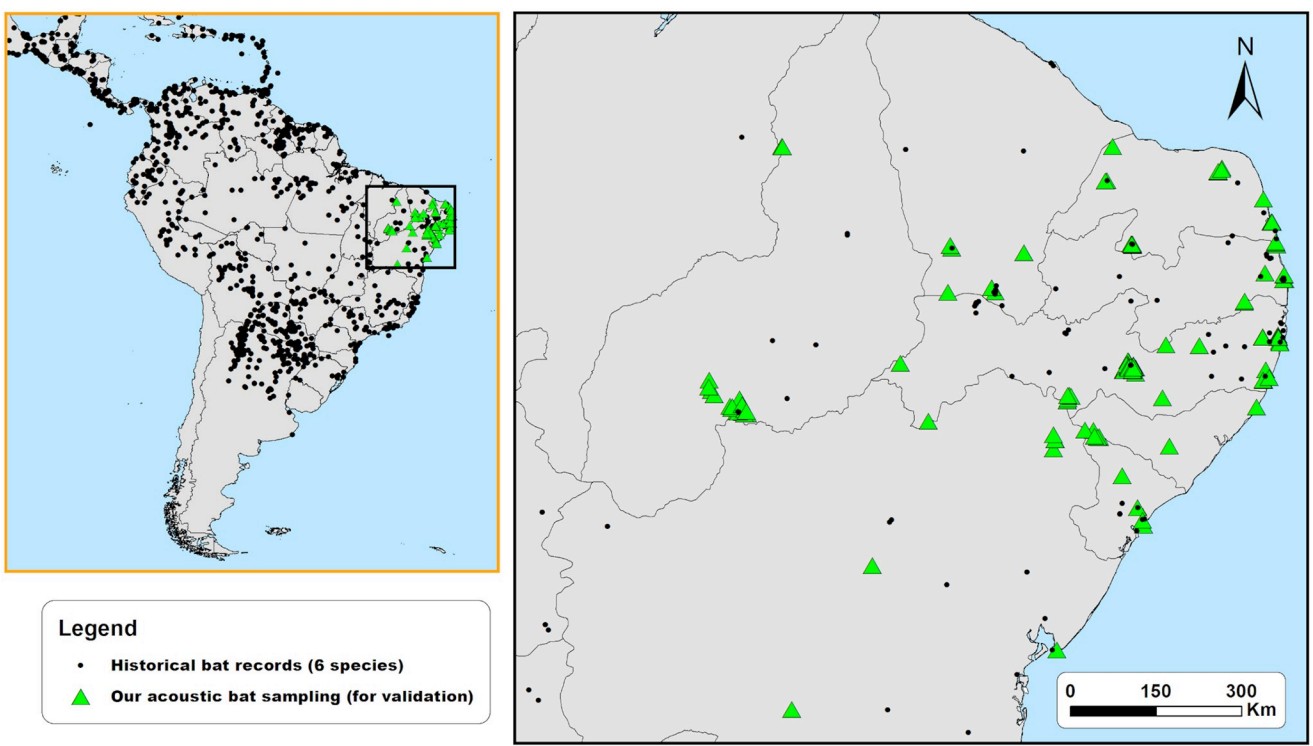

**Fig 1. Historical distribution records assembled from the literature review and the 129 acoustic sampling points used for the validation of SDM of six neotropical bat species (*Noctilio leporinus*, *Promops centralis*, *Promops nasutus*, *Pteronutus gymnonotus*, *Pteronotus personatus*, and *Saccopteryx leptura*).** Made with Natural Earth. Free vector and raster map data @ naturalearthdata.com.

[3, 33]. Since default regularization values lead to overfitted models when using spatial filtering [29, 34], we calibrated the models with different regularization multiplier values (default 1.0, 2.0, 3.0, and 4.0) [29, 35]. The regularization multiplier is a 'tunning' parameter used to smooth the distribution prediction of the model, making it more regular and less overfitted [29, 34]. To generate overall predictive distribution models, we used 75% of the data for calibration and 25% for internal evaluation (testing data).

We used ten cross-validation replications to test all models and calculate confidence intervals, resulting in 40 models for each species, and a total of 240 continuous suitability models for all species. Since we aimed to use bioacoustical data to validate binary maps (presence-absence), we used three widely used thresholds to convert the continuous suitability values of each model into binary maps: lowest presence threshold (hereafter LPT) [28]; 10th percentile of the predicted values (hereafter P10) [28]; and maximum sum of sensitivity and specificity (hereafter maxSSS) [36, 37]. LPT is considered the least conservative threshold, whereas maxSSS the most [36, 38]. We produced a total of 720 binary models (i.e., 120 binary models for each species: 6 species x 120 binary models).

Using MaxEnt's background and sample predictions for each model, we then evaluated each 120 species binary models for their predicting performance using four threshold-dependent theoretical evaluation metrics: (1) overall accuracy (hereafter OAcc) [39, 40]; (2) Cohen's maximized kappa statistics (hereafter P-kappa) [41, 42]; (3) True Skill Statistics (hereafter TSS) [40, 43]; and, (4) Symmetric Extremal Dependence Index (hereafter SEDI) [44]. OAcc measures the model predicted accuracy using the rate of correct classifications (true positive + true negative) and ranges from 0 to 1; while P-kappa, TSS, and SEDI measure the model predicted

accuracy taking into account the predicted accuracy of a by-chance model [40, 44]. P-kappa, TSS, and SEDI range from -1 to +1, where values ≤ 0 suggest a performance equal or worse than random, and as close the values get to +1, the better the prediction [40, 44]. For the math behind each theoretical evaluation metrics calculation, please consult Allouche, Tsoar [40], Cohen [41], Peirce [43], and Wunderlich, Lin [44].

As expected, the theoretical metrics did not converge on the same best models, therefore we selected the two best-scored binary models based on each metric (i.e., the two best-scored based on OAcc, the two best-scored based on P-kappa, and so on. . .) for each of the thresholds used (LPT, P10, and maxSSS). Then, we validated in the field the resulting 24 theoretical best binary models for each species, 144 binary models for the six studied species.

## Acoustic monitoring and species identification

For the selection of the sampling points for field validation, we summed the MaxEnt's given average potential distribution outputs of the six species to identify regions with the highest and lowest suitability of species occurrence, but without historical records. Considering those species' potential distribution covered extensive areas, we focused our field validation on 129 randomly-selected sampling points along an area of 758,193 km$^2$, in the Northeastern part of Brazil (Fig 1, and S3 Table). Since some Northeastern Brazil areas are not easily accessed, we pre-imposed the point selection near roads or paths accessible by, at least, an off-road vehicle.

Between March 2014 and January 2020, we employed passive acoustic monitoring to sample bat echolocation calls in the 129 sampling points (Fig 1, and S3 Table), using a combination of two SM2Bat+, two SM3BAT, and two SM4BAT-FS ultrasound recorders (Wildlife Acoustics Inc., Massachusetts, USA). We set the microphones at 45º to the ground, avoiding highly cluttered areas [16, 45, 46]. Since the highest frequency used by the studied species is ~60 kHz (*Noctilio leporinus*) [19], we configured the bat detectors with a minimum sampling rate of 384 kHz and 16 bit audio depth, enough to detect and record our focal species without distortions (e.g., aliasing). Each sampling point was acoustically monitored for at least two nights (from 2 to 15 nights), recording continuously from 30 minutes before sunset until 30 minutes after sunrise. The recordings were stored automatically in mono.wav format in the SD cards with a preset maximum duration of 1 minute if any sound above 7 kHz exceeded at least 6 dB. Since bat activity and the reception of the calls can be affected by weather and local conditions, we sampled only during nights with temperature > 15ºC, without strong winds (< 5 m/s) or rain [16, 45, 46].

We used Raven Pro 1.5 (The Cornell Lab of Ornithology 2014) for the acoustic analysis. To ease the acoustic analysis, we were divided the raw 1-minute sound files into 15-sec files and, subsequently, we visually inspected all files for the desired bat species calls after configuring the spectrograms to DFT equals 1024, 96% overlap, window length to 1 ms, using Hamming windows. We only analyzed sequences containing a minimum of three search calls with a good signal-to-noise ratio (> 15 dB) [47, 48]. We performed manual acoustic identification using qualitative (e.g., call structure and modulation) and quantitative parameters (e.g., frequency of maximum energy, maximum and minimum frequency, call duration, etc.), following previously published studies on neotropical bat acoustic identification [e.g., 18–20, 48–50]. We did not use feeding-buzzes (and calls immediately before and after) or social-calls for identification purposes. Although the chosen species have different natural histories, all are considered as common in the sampled region and can be easily detected and identified by their calls. All fieldwork procedures complied with the American Society of Mammalogists´ guidelines for the use of wild mammals in research and education [51] and were previously approved by the Brazilian Ministry of the Environment (SISBIO n.º 59743–1).

### Field validation of the models

We evaluated the total 144 selected binary models for the six focal species against the results from the acoustic monitoring performed in the field. We used a confusion matrix to compare the accuracy of the binary maps, where the observed presence and absence cases from the acoustic monitoring were compared against the predicted presence and absence of the models. This procedure allowed us to quantify true positives (TP), true negatives (TN), false positives (FP; commission errors, type I errors), and false negatives (FN; omission errors, type II errors).

We used six metrics for the model performance evaluation: (1) Accuracy, to quantify how often the model is correct in the overall prediction [Accuracy = (TP + TN) / total cases]; (2) Precision, to quantify how often is the model correct when it predicts the occurrence of the species [Precision = TP / (TP + FP)]; (3) Sensitivity (true positive rate, or recall), to quantify the ability of the model to predict species occurrence [Sensitivity = TP / (TP + FN)]; (4) Specificity (true negative rate), which quantifies the ability of the model to predict species absence [Specificity = TN / (FP + TN)]; (5) geometric mean of sensitivity and specificity (g-mean) is a performance metric for imbalanced classifications, with high g-mean indicating a right balance between sensitivity and specificity [g-mean = $\sqrt{}$ (Sensitivity * Specificity)]. If the species presence classification performance is weak, the g-mean will be low even with an excellent species absence classification performance [52]; and (6) harmonic mean of precision and sensitivity [f-score = (2 * Precision * Sensitivity) / (Precision + Sensitivity)], gives the same importance to precision and sensitivity, i.e., high F-score indicates excellent model performance on the minority class [52, 53]. The commission error rate is inversely proportional to sensitivity (= 1 —sensitivity), whereas the omission error rate is inversely proportional to specificity (= 1— specificity). The Fig 2 presents a flowchart summarizing the methodology applied for the modeling and model validation presented in this study.

To assess the overall performance of the theoretical evaluation metrics (OAcc, P-kappa, TSS, and SEDI), we used the R package 'lmerTest' [54] to perform a mixed-effects linear models to evaluate the correlations between those scores and the post-validation performance metrics scores obtained using acoustic monitoring (accuracy, precision, sensitivity, specificity, g-mean, and f-score). Species were treated as random effect factor. To test differences in the prediction performances (using the post-validation performance metrics scores) between the thresholds used (LPT, maxSSS, and P10), we employed the Kruskal-Wallis test with Mann-Whitney pairwise *post hoc* test. The thresholds were tested with both all species together and separately.

## Results

### SDMs and acoustic field validation

The acoustic sampling performed in this study resulted in more than 1.5 TB of raw 1-minute sound files. Those raw sound files were divided in 15-sec files and subsequently found that more than 300,000 of those 15-sec sound files contained bat calls. As shown in the Fig 3, the calls of the six studied species are very conspicuous and easily to identify. We identified echolocation calls of *Noctilio leporinus* in 38 points (29,4%), of *Promops centralis* in 23 (17,8%), *Promops nasutus* in 44 (34,1%), *Pteronotus gymnonotus* in 53 (41,1%), *Pteronotus personatus* in 21 (16,3%), and *Saccopteryx leptura* in 24 of the 129 sampled points (18,6%) (S3 Table). Before validation, and as expected, we found that the threshold choice on the models have a great influence on the predicted species occurrence areas as the Fig 4 exemplifies for one of the models generated for species *Promops nasutus*.

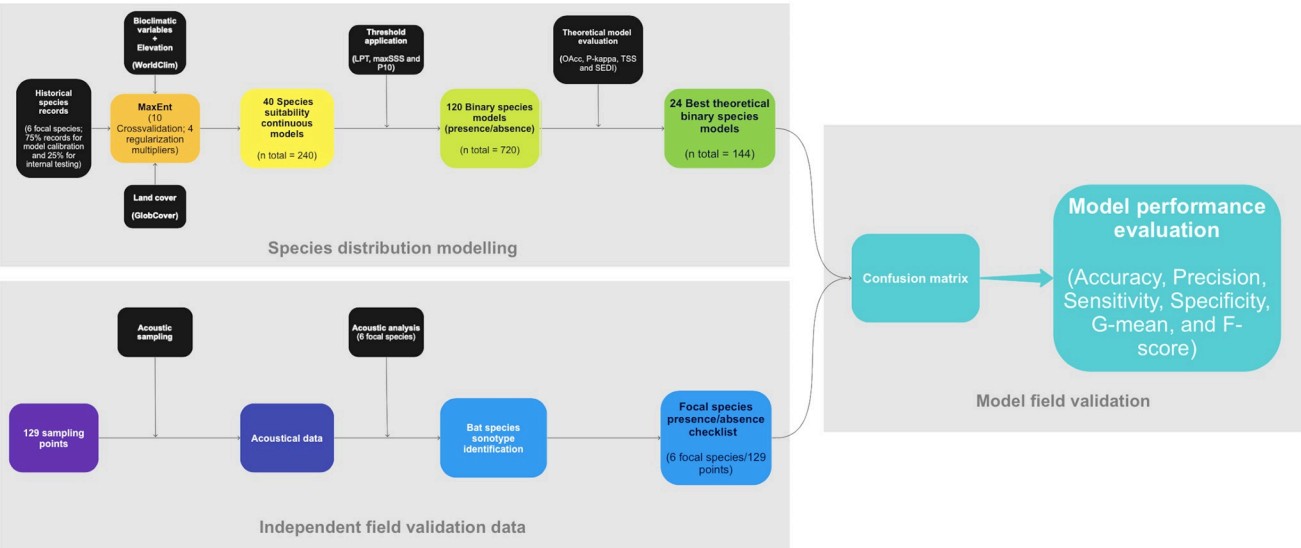

**Fig 2. Flowchart summarizing the methodology applied for the modeling and model validation for the six neotropical bat species presented in this study.**

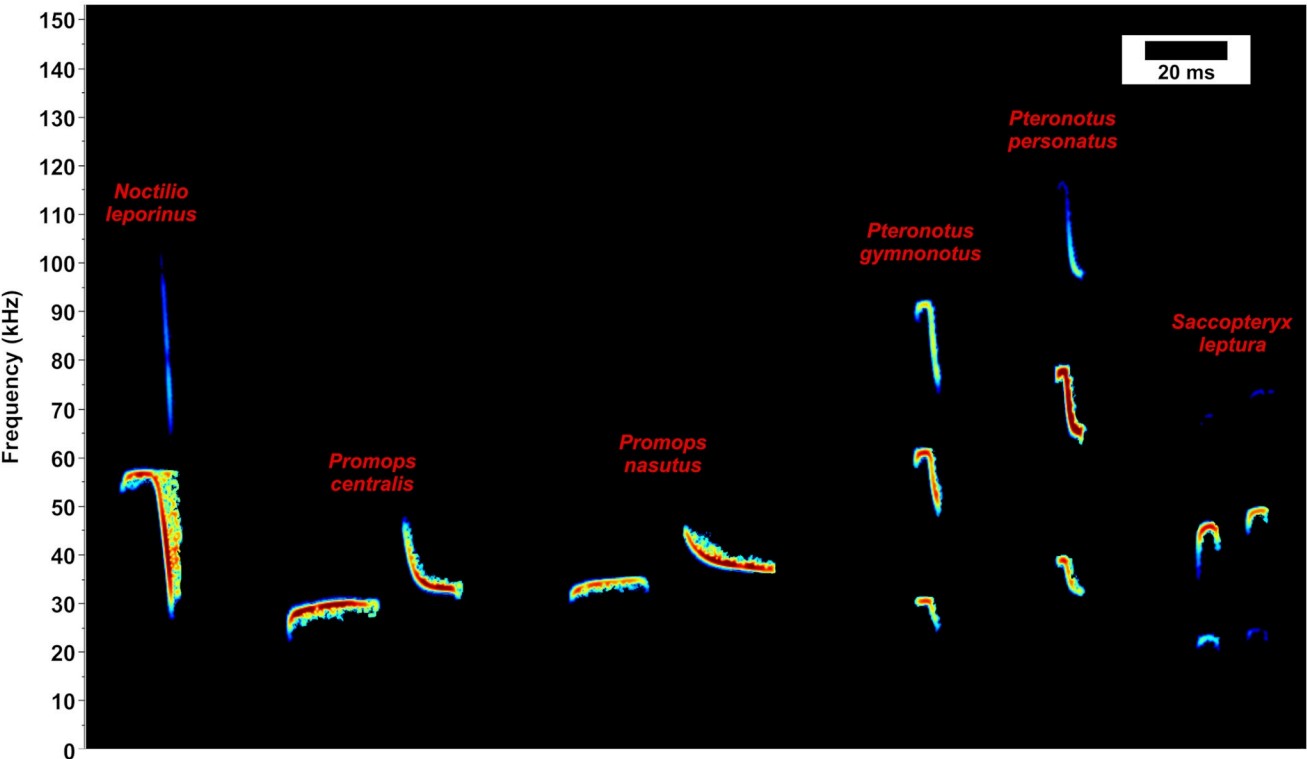

**Fig 3. Spectrogram of the echolocation calls from six neotropical bat species of this study (*Noctilio leporinus*, *Promops centralis*, *Promops nasutus*, *Pteronotus gymnonotus*, *Pteronotus personatus*, and *Saccopteryx leptura*).** Time scale (20 ms) in the upper right corner of the figure.

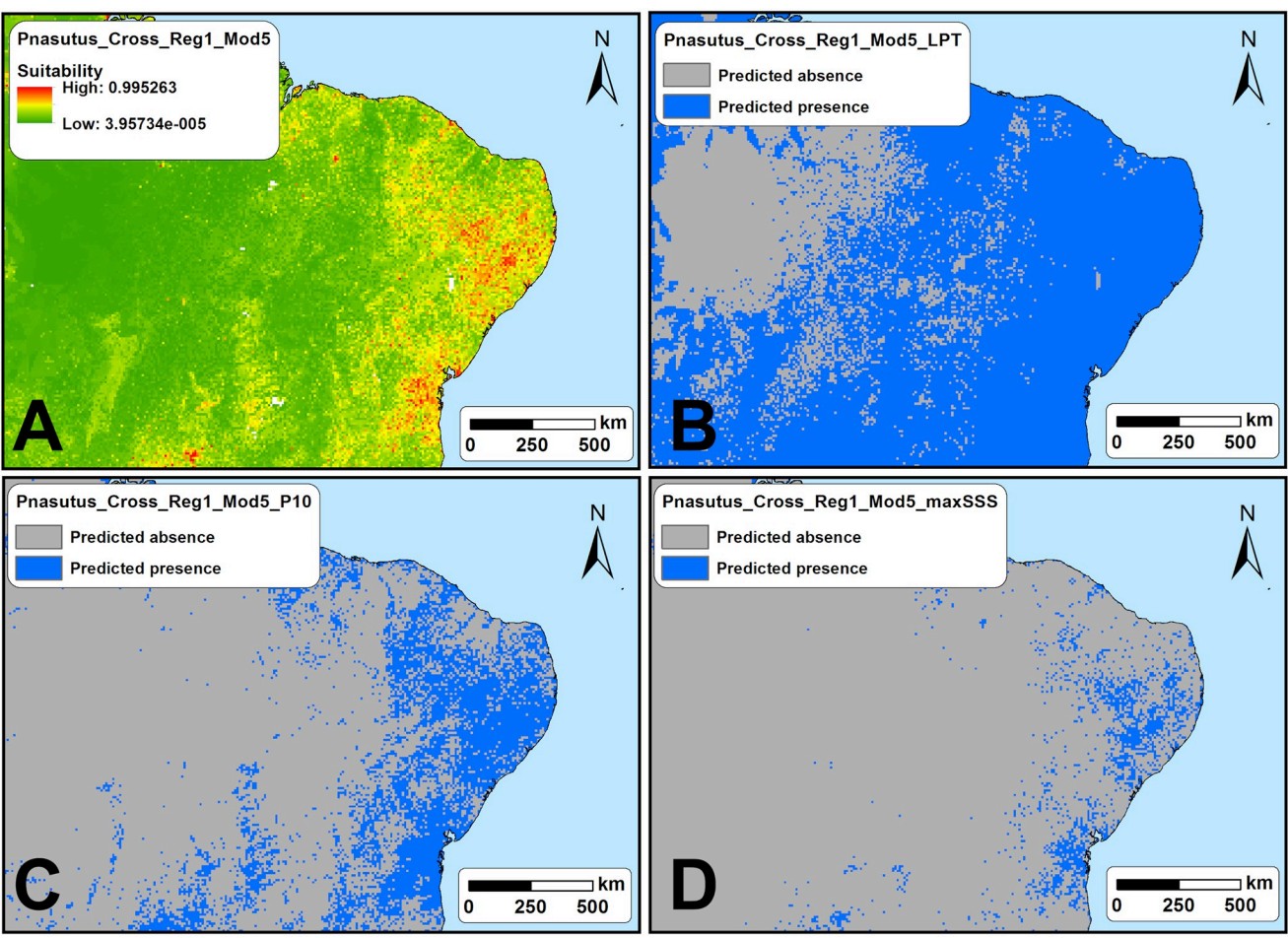

**Fig 4. Maps from a model (Pnasutus_Cross_Reg1_Mod5) generated for species *Promops nasutus*.** (A) continuous suitability model; (B) LPT-thresholded binary map (predicted presence-absence); (C) P10- thresholded binary map (predicted presence-absence); (D) maxSSS-thresholded binary map (predicted presence-absence). Made with Natural Earth. Free vector and raster map data @ naturalearthdata.com.

After field validation, the performance scores varied considerably between the 144 binary models: accuracy varied from 0.16 to 0.81, precision varied between 0.09 and 0.59, sensitivity varied from 0.17 to 1, specificity from 0 to 0.86, g-mean from 0 to 0.75, and f-score from 0.12 to 0.60 (S4 Table). We registered the highest accuracy score (0.81) in a maxSSS thresholded map of *Saccopteryx leptura*, the highest precision score (0.59) in a maxSSS thresholded map of *Noctilio leporinus*, and the highest sensitivity score (= 1) in LPT thresholded maps of four species (*Noctilio leporinus*, *Pteronotus gymnonotus*, *Promops nasutus*, and *Pteronotus personatus*) (S4 Table and S1 File). The highest specificity score (0.86) was recorded in maxSSS thresholded maps of two species (*Noctilio leporinus* and *Saccopteryx leptura*), the highest g-mean score (0.75) in a P10 thresholded map of *Saccopteryx leptura*, and the highest f-score score (0.60) in a maxSSS thresholded maps of *Pteronotus gymnonotus* (S4 Table and S1 File).

## Model evaluation vs. field validation

All theoretical model evaluation metrics analyzed exhibited a significant monotonic positive correlation with accuracy and specificity (Table 1). However, the evaluation metrics exhibited an overall significant negative correlation with sensitivity. We also found weak positive

**Table 1. Mixed-effects linear models results of the theoretical evaluation metrics and the post-validation performance metric scores for the distribution modelling of six neotropical bats based on bioacoustic field validation in northeastern Brazil.**

| Theoretical evaluation metric | Validation performance metric | Estimate | Std. Dev. | df | t-value | p-value |
|---|---|---|---|---|---|---|
| TSS | Accuracy | 0.57591 | 0.08081 | 141 | 7.127 | < 0.001 |
| | Precision | 0.71321 | 0.19701 | 106 | 3.62 | < 0.001 |
| | Sensitivity | -0.39601 | 0.05979 | 138 | -6.623 | < 0.001 |
| | Specificity | 0.36451 | 0.04473 | 139 | 8.15 | < 0.001 |
| | G-mean | 0.50775 | 0.06001 | 139 | 8.46 | < 0.001 |
| | F-score | -0.06236 | 0.15176 | 71 | -0.411 | n.s. |
| OAcc | Accuracy | 0.94031 | 0.06659 | 139 | 14.122 | < 0.001 |
| | Precision | 1.22879 | 0.20580 | 129 | 5.971 | < 0.001 |
| | Sensitivity | -0.74413 | 0.04034 | 139 | -18.45 | < 0.001 |
| | Specificity | 0.62894 | 0.02697 | 137 | 23.32 | < 0.001 |
| | G-mean | 0.72251 | 0.05290 | 139 | 13.659 | < 0.001 |
| | F-score | -0.29126 | 0.14502 | 29 | -2.008 | n.s. |
| P-kappa | Accuracy | 0.18566 | 0.04241 | 141 | 4.378 | < 0.001 |
| | Precision | 0.22173 | 0.09465 | 81 | 2.343 | < 0.05 |
| | Sensitivity | -0.11143 | 0.03156 | 138 | -3.53 | < 0.001 |
| | Specificity | 0.11323 | 0.02429 | 140 | 4.662 | < 0.001 |
| | G-mean | 0.18451 | 0.03183 | 140 | 5.797 | < 0.001 |
| | F-score | 0.02633 | 0.07437 | 86 | 0.354 | n.s. |
| SEDI | Accuracy | 0.24738 | 0.10510 | 94 | 2.354 | < 0.05 |
| | Precision | -0.006723 | 0.222993 | 73 | -0.03 | n.s. |
| | Sensitivity | -0.39010 | 0.07094 | 90 | -5.488 | < 0.001 |
| | Specificity | 0.23899 | 0.05936 | 92 | 4.026 | < 0.001 |
| | G-mean | 0.22122 | 0.09098 | 93 | 2.431 | < 0.05 |
| | F-score | -0.51451 | 0.17516 | 94 | -2.937 | < 0.001 |

df, degreeds of freedom; n.s., not significative.

correlation between P-kappa and precision, and a very weak negative correlation between SEDI and precision. Although negative, the correlations between the majority of theoretical evaluation metrics and f-score were not significant, except for SEDI that displayed a significative moderate negative correlation with f-score. Only TSS and overall accuracy exhibited significant moderate or strong positive correlations with g-mean.

## Thresholds vs. validation

Based on the output maps of all species together, LPT thresholded predictions exhibited significantly the lowest overall averaged accuracy, precision, specificity, and g-mean scores of the three thresholds tested (Fig 5, and S5 Table). While P10-based models exhibited a significantly higher averaged f-score than maxSSS-based, we found no significant differences in f-score between P10 and LPT, and between maxSSS and LPT-based models (Fig 5, and S5 Table). P10-based models obtained overall averaged sensitivity scores higher than maxSSS but lower than LPT-based models and specificity scores higher than LPT but lower than maxSSS-based models (Fig 5, and S5 Table). LPT-based predictions scored the highest averaged sensitivity but also presented low average specificity scores. Note that while LPT's sensitivity scores near one, its specificity scores are also near zero (Fig 5). We found this same behaviour on LPT and maxSSS-based models when we analyzed all species separately (Fig 5, and S5 Table). One LPT prediction for *N. leporinus* (Fig 6) exemplifies this odd behavior, where omission errors are

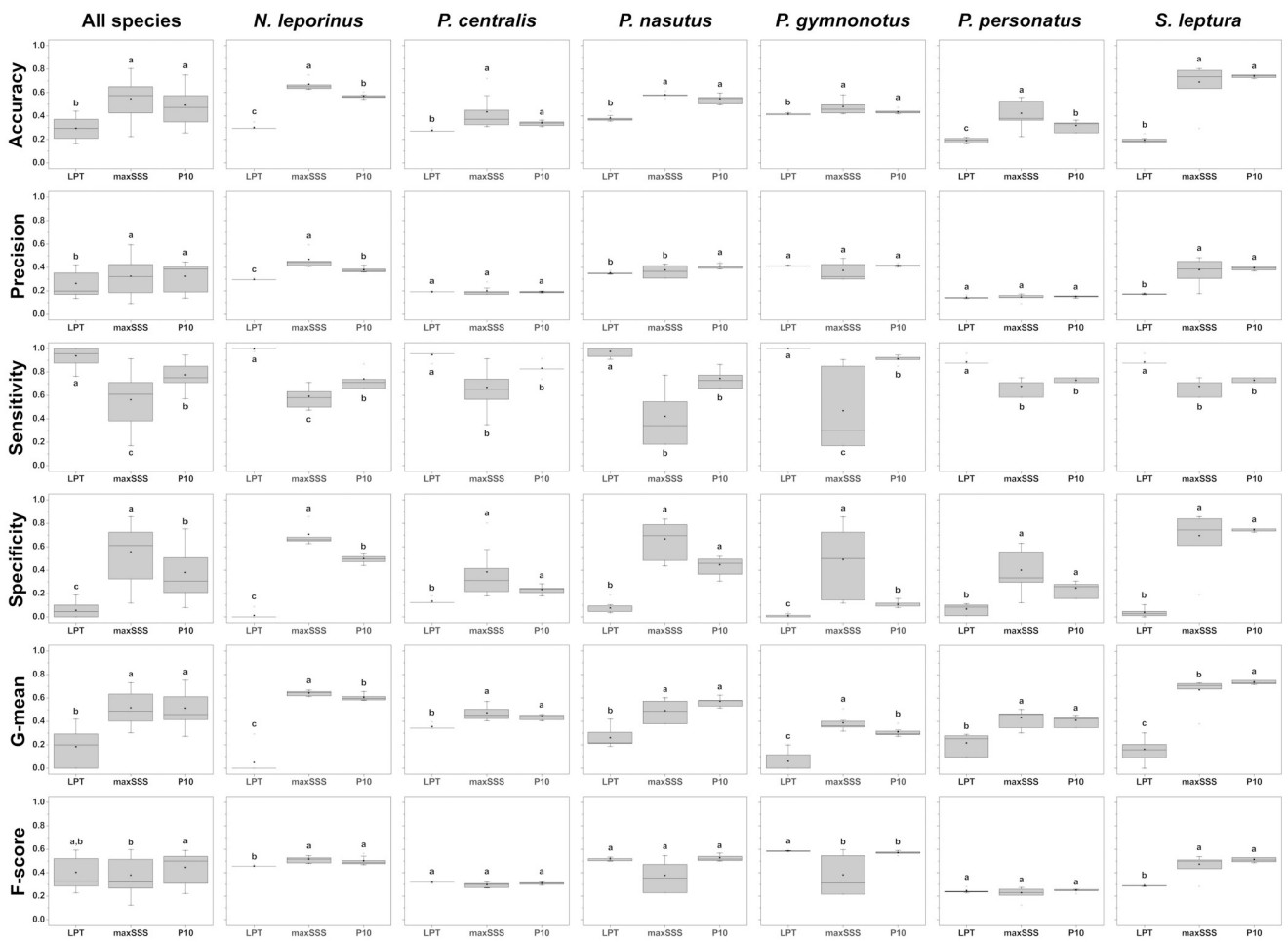

**Fig 5. Box-plots of the accuracy, precision, sensitivity, specificity, g-mean, and f-score scores of LPT, maxSSS, and P10 thresholded maps.** Lower and top box boundaries represent the 25th and 75th percentile, respectively. The point inside the box represents the average, while the line represents the median. Error lines represent 1.5*interquartile range, and any asterisk outside the error lines represent outliers. Different letters indicate significant differences between groups (p < 0.05).

minimal, but commission errors are maximum (sensitivity = 1, specificity = 0, and g-mean = 0). Even the LPT prediction with the highest g-mean score (*P. nasutus*, Fig 7) represented a commission error rate of ~81% and an omission error rate of ~7% (sensitivity = 0.93, specificity = 0.19, and g-mean = 0.42). While maxSSS-based models scored the highest overall averaged accuracy and specificity, it also exhibited the lowest averaged sensitivity scores (S5 Table). Nevertheless, in contrast to LPT, maxSSS-based models' sensitivity and specificity averaged scores are similar, thus presenting higher g-mean scores than LPT (Fig 5, and S5 Table). This was clear when the maxSSS-based models with the highest g-mean (Fig 8) represented an omission error of ~29% and a commission error of ~25% (sensitivity = 0.71, specificity = 0.75, and g-mean = 0.73). All thresholds significantly presented different specificity and specificity scores, but we found no significant differences between the overall accuracy, precision, and g-mean scores of maxSSS and P10 thresholded predictions (Fig 5, and S5 Table).

Analyzing species by species, LPT-based predictions once again significantly exhibited the highest averaged sensitivity scores. However, it also showed significantly the lowest averaged accuracy, specificity, and g-mean scores for all six species studied (Fig 5, and S5 Table). In the

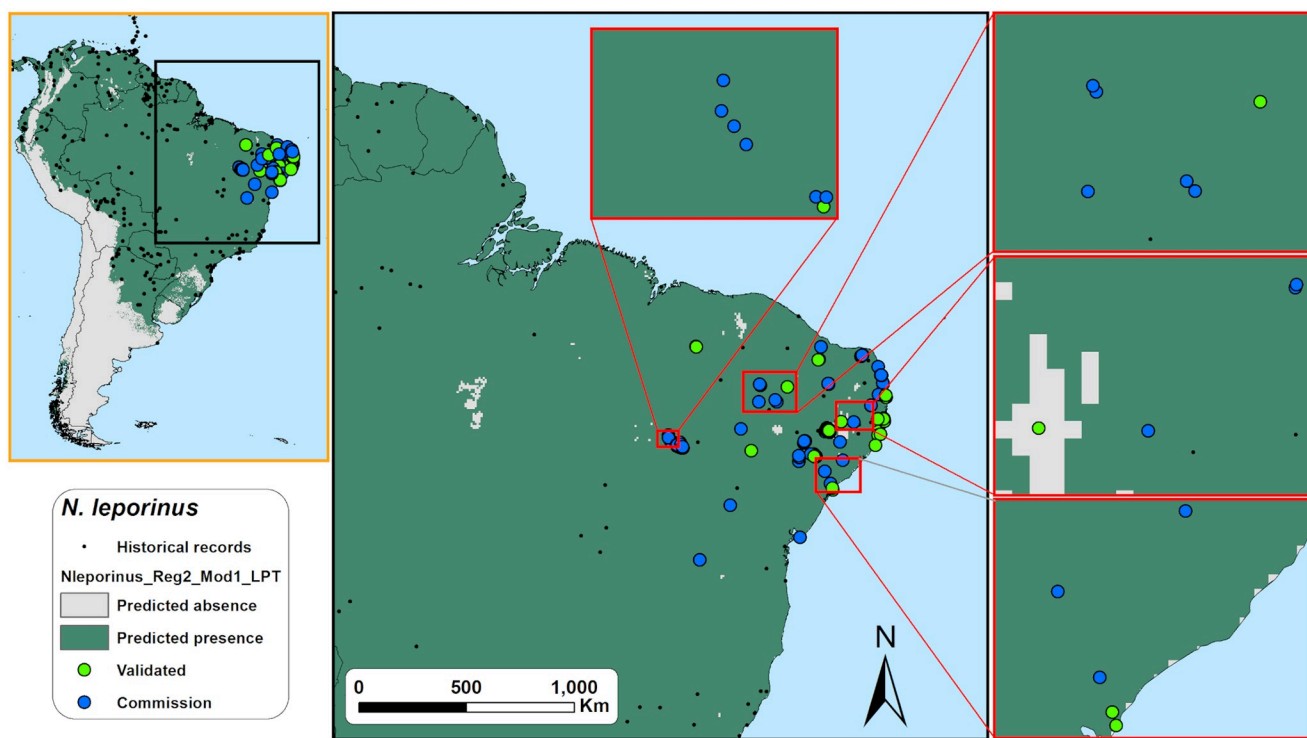

**Fig 6. Field validation results for the LPT binary map with the highest accuracy score (= 1) for *Noctilio leporinus* in northeastern Brazil.** Omission errors are minimal but commission errors are maximum (sensitivity = 1, specificity = 0, and g-mean = 0). 'Omission' points represent locations where the model did not predict the species occurrence, but the species was detected during the acoustics monitoring; 'Validated' points represent locations where the model predict the species occurrence and the species was detected during the acoustics monitoring or locations where the model did not predict the species occurrence, and the species was detected during the acoustics monitoring; 'commission' points represent locations where the model predict the species occurrence but the species was not detected during the acoustics monitoring. Made with Natural Earth. Free vector and raster map data @ naturalearthdata.com.

cases of *N. leporinus* and *S. leptura* (Figs 5–8), LPT-based predictions also significantly exhibited the lowest averaged precision and f-score, and the lowest averaged precision scores for *P. nasutus* and *P. personatus* among the three thresholds tested (Figs 5 and 7 and S5 Table). Thresholded predictions based on maxSSS significantly exhibited the lowest averaged sensitivity scores among the three thresholds tested for *N. leporinus* and *P. gymnonotus* (Fig 5, and S5 Table). Still, maxSSS-based predictions also showed significantly the highest averaged specificity and g-mean scores for the same species. The maxSSS-based predictions also presented the highest averaged accuracy for *N. leporinus* and *P. personatus*, and the highest averaged g-mean for *S. leptura* among the three thresholds tested (Fig 5, and S5 Table). *Saccopteryx leptura*'s P10 predictions significantly exhibited the highest averaged g-mean of the three thresholds tested (Fig 5, and S5 Table). We found no differences between the performance scores based on maxSSS and P10 in all *P. centralis* and *P. nasutus* predictions (Fig 5). We also found no differences between the three thresholds in precision and f-score for *P. centralis* and *P. personatus*, f-score of *P. nasutus*, and precision scores of *P. gymnonotus* maps (Fig 5). See S1 File for the best performing binary predictions for the six species.

## Discussion

This study evaluated and validated species binary distribution models using a combination of acoustic data collected in the field and simple performance metrics. We proved that

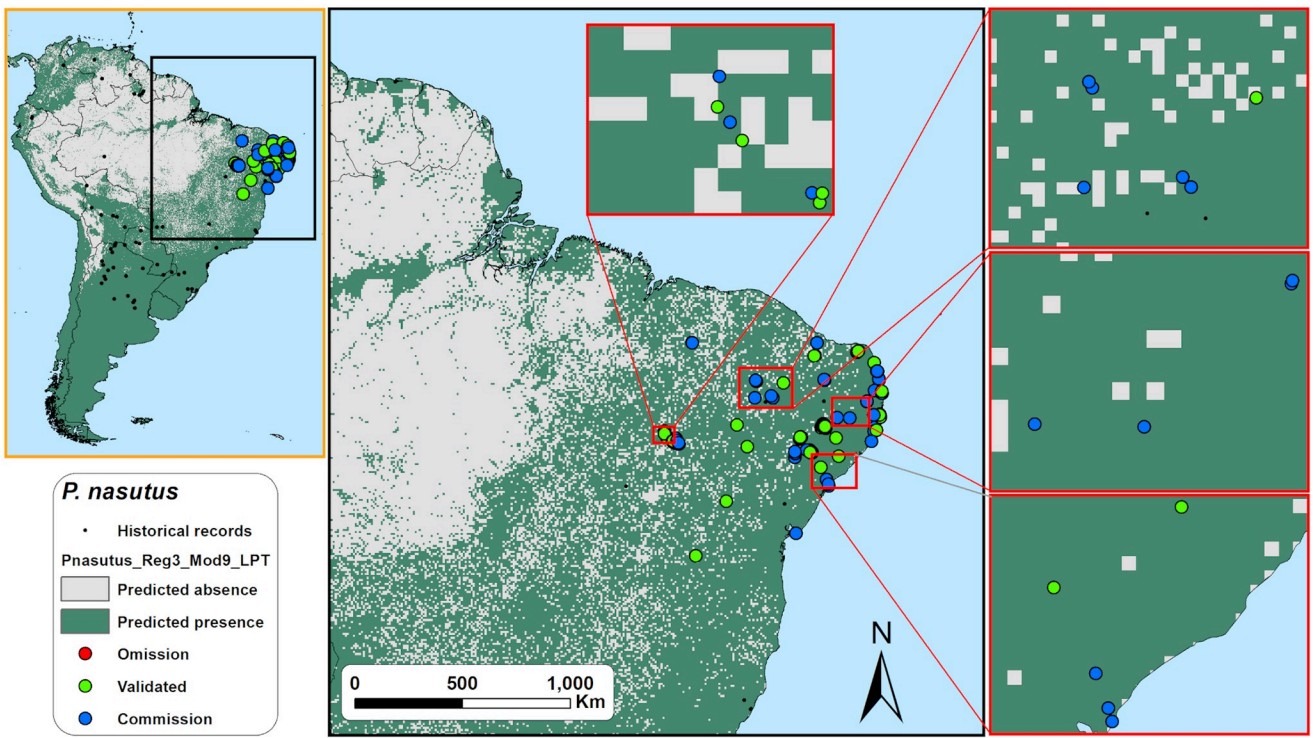

**Fig 7. Field validation results for the LPT binary map with the highest g-mean score (= 0.42) for *Promops nasutus* in northeastern Brazil.** Omission errors are low but commission errors are very high (sensitivity = 0.93, specificity = 0.19). See Fig 12 caption for the explanation on omission, validation and commission points. Made with Natural Earth. Free vector and raster map data @ naturalearthdata.com.

bioacoustics are a very effective method for the *in situ* validation of SDM, here we tested it for six neotropical bat species in a large and poorly-sampled area in Brazil. Species distribution modeling is subject to interpretation and depends on many factors, such as tuning parameters and thresholds [7, 8, 29, 35, 36, 38]. After using field validation, we urge modelers to explore their modelling approaches effects on result models since predictions might be very variable even when applying optimal practices.

After modelling the distribution of six bat species, we used acoustical samplings to confirm the predicted occurrence and absences forecasted by 144 distribution potential models. For species with conspicuous vocalizations—like most bat families—the acoustical samplings we employed showed the potential to better refine SDMs in large and under-sampled regions, with relatively low sampling effort. This is quite useful in tropical areas, usually rich in bat species, but frequently understudied [55, 56]. Using bioacoustics as a validation method, we also demonstrated that a careful decision on the modelling parameters and thresholds used is pivotal since, depending on the combination, they can produce very different outputs. Our observations highlight the importance and necessity of *in situ* validation of SDM outputs.

Field validation of SDM is unusual [e.g., 57, 58], and very rare for bats [e.g., 10, 59, 60]. Still, we empirically demonstrated that independent field surveys are the best approach to corroborate the predictions made by modelling, especially in subsampled regions with high biodiversity like the Neotropics. We demonstrated that only using pre-existing data subsets and theoretical evaluation metrics (TSS, overall accuracy, P-kappa, and SEDI), our model evaluation may be restricted to its particularities and constraints. Simply selecting the model with the highest score may not mean that we are choosing the model that best represents reality. For

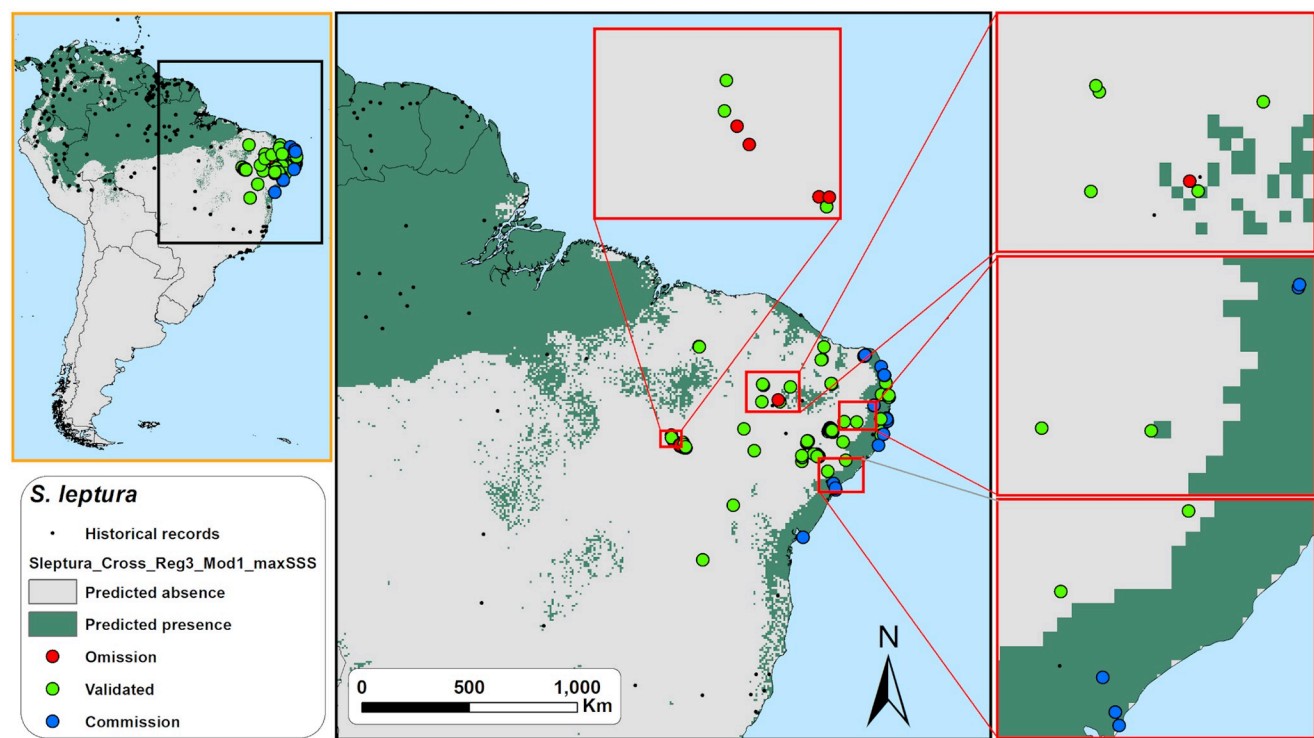

**Fig 8. Field validation results for the maxSSS binary map with the highest g-mean score (= 0.73) for *Saccopteryx leptura* in northeastern Brazil.** Omission errors and commission errors are balanced (sensitivity = 0.71, specificity = 0.75). See Fig 12 caption for the explanation on omission, validation and commission points. Made with Natural Earth. Free vector and raster map data @ naturalearthdata.com.

example, a maxSSS-based model (Nleporinus_Cross_Reg1_Mod4_maxSSS) obtained the best TSS score for *Noctilio leporinus* but also displayed very high omission errors (>30%) after we validated it on the field. Although all theoretical evaluation metrics correlated positively with specificity (true negative rate) and g-mean (good balance between specificity and sensitivity), they also correlated negatively with sensitivity (true positive rate). Thus, high theoretical evaluation metric' scores such as the widely used TSS may guarantee reliable models but could not provide the best model for our purposes and may also tend to score higher for the most over-fitted models. This problem can be exacerbated when dealing with species with fewer existing data records, such as *Promops centralis*, and highlights how crucial *in situ* validation of SDMs may be.

We are aware that *in situ* validation of SDM is not always possible, as that will depend on the focal species, its extension of occurrence, survey methods, and the type and accessibility of the potential area modelled. In case of the echolocating bats, acoustic surveys can provide a reliable and fast validation method for distribution models. However, field validation of SDM —in smaller or focally-selected parts of the predicted distribution, or randomly selected regions—should be imperative. This is especially important in a conservation-focused scenario dealing with such high habitat changes due to anthropogenic causes. Modelling species distributions without proper *in situ* validation may result in inaccurate outputs, compromising the implementation of better conservation policies or species management plans, for example [5, 9, 60, 61]. This can be particularly serious in the case of models with actual low sensibility (high omission errors).

## Theoretical model evaluation metrics and thresholds vs. validation

We found that all theoretical model evaluation metrics studied here correlated positively with accuracy. However, caution is necessary, since the most detected species (*P. gymnonotus*) was recorded in only 41% of the sampled points (meaning an unbalanced class data, i.e., in this case, absences are higher than presences). In situations like this, any random model predicting more absences than presences would be benefited by an evaluation metric that does not take into account results by chance—this is the case of accuracy [62]. Therefore, accuracy should not be used when data used to train and/or test the models is unbalanced. Here, we empirically demonstrated that sensitivity-specificity and precision-sensitivity metrics, as g-mean and f-score, are better performance measures for the SDMs evaluation than accuracy [52, 62]. For example, the *Pteronotus gymnonotus'* distribution output with the highest accuracy also had the third-highest omission rate. In opposition, the model with the highest g-mean and f-score also presented low omission scores. These results were not threshold-related since they occurred in two different maxSSS models, evidencing the unbalanced nature of our SDM outputs and the problem of using accuracy to measure model performance. Considering the majority of SDMs are based on unbalanced data, instead of accuracy, the use of sensitivity-specificity and precision-sensitivity metrics should be mandatory to test the models as they are not affected by unbalanced class data sets [52, 62].

Surprisingly, we also found that all theoretical threshold-dependent evaluation metrics tested here exhibited an overall significant negative correlation with sensitivity and a significant positive correlation with specificity. This means that models with higher evaluation scores predicted better locations with actual species absence than species presence. West, Kumar [63] reported similar findings after field validation of MaxEnt's invasive cheatgrass species models. This is probably because bioclimatic variables' values are more homogeneous in species presence locations than in absence. Nevertheless, this 'issue' will be less a concern if the modeler's goal is to balance actual presences and absences, as we found positive correlations between evaluation metrics and g-mean or f-score. So, how about the ability of four threshold-dependent theoretical evaluation metrics in predicting models' performance? In conclusion, the four theoretical model evaluation metrics studied here presented different performances about their capacity to predict the performance of models. Given the accuracy drawbacks when unbalanced data is in the game, metrics such as g-mean and f-score gain greater relevance than accuracy itself. Thus, TSS and overall accuracy (OAcc) obtained a higher ability to predict the performance of models regarding the balance between sensitivity and specificity (g-mean), however OAcc was the second-worst in terms of f-score. Still, given the strong positive correlation of most theoretical model evaluation metrics concerning specificity and negative concerning sensitivity, there is a great tendency of these metrics to better evaluate models with high omission rates. In conclusion, the choice of theoretical evaluation metrics to evaluate the models' performance might have a great impact on its selection, and it needs to be chosen carefully towards the proposed objective.

And how about the ability of the three studied thresholds to convert continuous species habitat suitability models into presence-absence maps? We found that threshold performances varied largely. Despite having almost no omission errors, the LPT models exhibited higher commission error rates and lower accuracy, g-mean, and precision scores. Thus, at least for the six widespread neotropical bat species studied here, we were able to empirically confirm Liu, White [37]'s findings: maps based on the LPT threshold are unsuitable for species distribution modelling. In our study, LPT-thresholded maps of the two species with the most historical records (*N. leporinus* and *S. leptura*) also had a worse performance than the other four LPT-modeled scores. Although widespread and common, *N. leporinus* is a piscivorous bat

species, strongly related to water bodies, and *S. leptura* is a forest-dwelling species that forage next to edges [64, 65]. Thus, these two species as less generalist than the other four we analyzed, and the use of a less conservative threshold can be detrimental in those cases. Therefore, knowledge of the species' natural history and the use of land cover data in the models might be fundamental for best SDM practices and better output results [66]. But contrary to Liu, White [37], we also found that some maxSSS maps with high accuracy scores were highly overfitted (exhibiting low sensitivity/high omission rates), sometimes excluding historical locations for some species. Nevertheless, we also found that some maxSSS and P10 thresholded maps performed reasonably well, exhibiting balanced results between sensitivity and specificity (displaying high g-mean scores). In conclusion, we believe LPT is unsuitable for species distribution modelling, because although it reduces the omission errors of the presence-absence maps to almost zero, it does so at the expense of high commission errors. However, even showing higher omission errors than LPT, P10 thresholded models performed best in predicting actual species occurrences, while maxSSS models performed best in predicting where we did not record the species in the field. Several authors agree that threshold selection (as other parameters) has a high impact on the binary map (presence/absence) outputs and the models' predictive capacity [e.g., 6, 8]. Several thresholds have been proposed and evaluated; however, most of those evaluations are based on theoretical evaluation metrics without independent field validation data. After using *in situ* validation in our study, we are cautionary about some studies still proposing to model species distribution using a single threshold.

Finally, we found that using arbitrary thresholds and theoretical evaluation metrics for modelling and evaluate the models' performance can be a precarious approach with many possible outcomes, even if getting good evaluation scores. Validating the models using a subset of the historical occurrence points is undoubtedly faster and less laborious than using independent data collected in the field. However, one cannot guarantee if the species are still present in historical points in databases such as GBIF [67]. Hence, using independent field data is the safest way to validate the species´ presence in the modelled region and should not be overlooked.

## Supporting information

**S1 Table. Location of the 1277 single records of the *Noctilio leporinus*, *Promops centralis*, *Promops nasutus*, *Pteronotus gymnonotus*, *Pteronotus personatus* and *Saccopteryx leptura*, gathered after our bibliographic revision.**
(XLSX)

**S2 Table. Prediction variables used in this study for the distribution modelling of six neotropical bat species in northeastern Brazil.**
(XLSX)

**S3 Table. Presence/absence obtained in the acoustic monitoring of 129 acoustic sampling points used for the validation of SDM's of six neotropical bat species in northeastern Brazil.**
(XLSX)

**S4 Table. Theoretical model evaluation scores, validation confusion matrix, and performance scores of the 144 binary distribution models of six neotropical bat species validated with field acoustics sampling in northeastern Brazil.**
(XLSX)

**S5 Table. Average ± standard deviation scores of accuracy, precision, sensitivity, specificity, g-mean, and f-score of the three thresholds tested for SDMs of six neotropical bat**

**species after a field validation in northeastern Brazil using bioacoustics.** Kruskal-Wallis test results for the comparisons between the thresholds' performance scores is also presented. The Mann-Whitney pairwise post hoc test results are presented by letters next to average ± standard deviation (different letters indicate significant differences between groups, p < 0.05).
(DOCX)

**S1 File. Field validation results for the binary maps with the highest accuracy, precision, sensitivity, specificity, gmean, and f-score scores for the six studied species in northeastern Brazil.** Maps made with Natural Earth. Free vector and raster map data @ naturalearthdata. com.
(PDF)

## Acknowledgments

We thank all those who helped during fieldwork. We are also grateful to ICMBio and the surveyed conservation areas' staff for all the support. FH thanks the Raven Team Bioacoustics Research Program for the free copy of Raven software. This manuscript was based on one of the chapters of FH's PhD Thesis at PPGBA/UFPE, and we thank the five doctoral examiners for their constructive comments during the defense.

## Author Contributions

**Conceptualization:** Frederico Hintze, Ricardo B. Machado, Enrico Bernard.

**Data curation:** Frederico Hintze, Ricardo B. Machado.

**Formal analysis:** Frederico Hintze, Ricardo B. Machado.

**Funding acquisition:** Enrico Bernard.

**Investigation:** Frederico Hintze.

**Methodology:** Frederico Hintze, Ricardo B. Machado.

**Project administration:** Frederico Hintze, Enrico Bernard.

**Resources:** Enrico Bernard.

**Supervision:** Enrico Bernard.

**Validation:** Frederico Hintze.

**Writing – original draft:** Frederico Hintze, Enrico Bernard.

**Writing – review & editing:** Frederico Hintze, Ricardo B. Machado, Enrico Bernard.

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
