## [Decision Letter · Decision Letter 0]

29 Apr 2021

PONE-D-21-07379

Bioacoustics for in situ validation of species distribution modelling: An example with bats in Brazil

PLOS ONE

Dear Dr. Hintze,

Thank you for submitting your manuscript to PLOS ONE. After careful consideration, we feel that it has merit but does not fully meet PLOS ONE’s publication criteria as it currently stands. Therefore, we invite you to submit a revised version of the manuscript that addresses the points raised during the review process.

We look forward to receiving your revised manuscript.

Kind regards,

Daniel de Paiva Silva, Ph.D.

Academic Editor

PLOS ONE

Journal Requirements:

Please ensure that you refer to Figure 9 in your text as, if accepted, production will need this reference to link the reader to the figure.

3. We note that Figures 1,9,10,11 and A-U in your submission contain map images which may be copyrighted. All PLOS content is published under the Creative Commons Attribution License (CC BY 4.0), which means that the manuscript, images, and Supporting Information files will be freely available online, and any third party is permitted to access, download, copy, distribute, and use these materials in any way, even commercially, with proper attribution. For these reasons, we cannot publish previously copyrighted maps or satellite images created using proprietary data, such as Google software (Google Maps, Street View, and Earth). For more information, see our copyright guidelines: http://journals.plos.org/plosone/s/licenses-and-copyright.

1.              You may seek permission from the original copyright holder of Figures 1,9,10,11 and A-U to publish the content specifically under the CC BY 4.0 license. 

Additional Editor Comments:

Dear Hintze et al.,

After two independent reviews, I believe your manuscript may be accepted for publication after a major review when you will have the possibility to take care of the issues raised by both reviewers. The reviewers decided on a major and a minor review. Although both reviewers raised important issues, please take special care regarding the issues raised by reviewer #2, who raised important issues related to your writing and requested you to be more explicit in the relation between the study's goals and results.

Considering the pandemic situation in Brazil, I believe a three-month (August 1st 2021) period will be more than enough for you to deliver the revised version of your text. By that time, along with the revised version of your text, please do not forget to prepare a rebuttal letter where you will explain all the decisions you took regarding the issues raised by the reviewers. Do not hesitate to resubmit earlier in case you are able to. Nonetheless, in case you need more time, please let me know.

Sincerely,

Daniel Silva

Reviewers' comments:

Reviewer's Responses to Questions

**Comments to the Author**

1. Is the manuscript technically sound, and do the data support the conclusions?

Reviewer #1: Yes

Reviewer #2: Partly

2. Has the statistical analysis been performed appropriately and rigorously? 

Reviewer #1: Yes

Reviewer #2: No

3. Have the authors made all data underlying the findings in their manuscript fully available?

Reviewer #1: Yes

Reviewer #2: Yes

4. Is the manuscript presented in an intelligible fashion and written in standard English?

Reviewer #1: Yes

Reviewer #2: Yes

5. Review Comments to the Author

Reviewer #1: Review for PONE-D-21-07379

In this manuscript, the authors use a sound presence-only database of neotropical bats to build presence-absence maps based on species distribution modeling (SDM) techniques, specifically using MaXent. The authors test distinct combinations of parameters and validate the dataset using an impressive independent dataset, obtained with acoustic monitoring. I appreciate the introduction and believe that sampling procedures and methodological procedures are adequate for the study. The results section seems to have some results missing, and the figures could have some more editing to facilitate reading the many findings of this study. Finally, I believe that the discussion is fine in content, but could be organized to improve interpretations, with a clear take-home message for the broad public. One last point is that although the title contains bioacoustics, I missed seeing some discussion on the cost-benefit of acoustic and non-acoustic sampling to validate SDM.

This diligent study will be a good contribution to the field. Bellow, I provide some suggestions and commentaries through the manuscript to be considered before publication.

Abstract

L. 15: estimates of species diversity?

Introduction

L. 41-46: some sentences are repeated from the abstract. Consider rephrasing them.

L. 55: Is there a spatial modeling science? Maybe spatial modeling, or just modeling, would be the method to make science (ecology) more applied, where errors should be minimized.

L. 74: I think the goal is to evaluate SDM performance given the validation dataset obtained with bioacoustics and different thresholds. Having four evaluation metrics would then be a methodological procedure.

Methods

Perhaps a flowchart with the steps and setting used to build models could be provided to have a clearer picture of the distinct models.

L. 131-132: could you provide, in few words, how does the regularization multiplier parameter work?

L. 167: The validation points seem to be more concentrated toward east, not randomly distributed across the 1000 x 1000 km area. This pattern is similar to the historical records, which may likely be associated with accessibility.

L. 181: Did the acoustic monitoring took place in all months from march 2014 – January 2020? Can you provide some information about the seasonal activity of these species?

L. 182: In line 71 you mention 300,000 files. Are these files subsets within the total continuous files? If so, please, describe how subset files were selected.

Results

* You could add a paragraph summarizing the main results found in SDM with the different settings used in the models, before validation.

* What are the differences in performance found for different regularization multiplier values?

L. 236: It would be nice to see an image of the distinct calls found in such an amazing acoustic dataset.

Figures: It would be easier to see the graphs if you provide a single panel summarizing all seven figures 2-8, where each line could be a metrics (accuracy, precision, etc.) for the distinct thresholds, and boxplots for each species would be different colors.

Discussion

* The first paragraph still lacks a clear summary of the main findings and a take-home message. For instance, it remains unclear what is behind the differences among the three thresholds used in this study. What are the fundamental differences between them, and which would theoretically provide a more informative result? It would also be interesting if you could discuss which models had best performance and if post-validation performance of these specific models had high and positive correlation.

* Correlation scores in table 1 are not strong, with most <0.5. This finding should be highlighted and discussed.

* Also, it would be interesting to know if the performance metrics are similar to studies that rely upon non-acoustic methods for validation.

L. 374-375: This is not entirely true for all species/thresholds evaluated and could be acknowledged here. In this first paragraph, you could be more specific in the take-home message. For instance, you could include information on how different thresholds may be better than others and discuss the influence of sample size and unbalanced data.

Reviewer #2: This study examines the contribution of bioacoustic tools (passive acoustic monitoring) to validate the predictions of Species Distribution Models (SDM) in six tropical bat species. Based on the comparison between theoretical evaluation metrics and post-validation performance parameters obtained from field sampling, the authors highlight the need of in situ validation of SDM and argue the use of novel acoustic techniques as rapid validation methods. The study has been properly conducted, using sound methods and a large data set, which enable the authors to successfully address the proposed goals. Overall, the manuscript is clear, well-written and presents results in an effective manner. Nevertheless, there are still a series of issues that should be carefully revised before publication. First, I strongly recommend an English revision of the whole text by a native speaker or language service, if it has not been made yet. I am not an English native speaker, but I feel this is needed to significantly improve spelling, grammar, and the general flow of the text. All across the manuscript, I included suggestions and minor questions (directly on the pdf; see attachment) that aim to increase the clarity and precision of the document. My major points are listed below.

Major points

1. Statistical analysis

My main concern is related to the statistical analyses, since some of them may be fell into pseudoreplication. As shown in Table 1, Spearman correlation tests were calculated using 144 observations (except for SEDI) that came from models of the six study species (24 models per species). Thus, these observations (validation metrics) are grouped by species and they must be not considered as independent replicates. Predictions obtained by models of the same species can likely be related. As consequence, the statistical analysis applied to examine the correlation between theoretical evaluation and in situ validation should be designed taking into account the non-independence of the observations and performed again. I recommend the use of general linear mixed-effects models (GLMM), for instance, with species as random factor. Probably, the same might be applied to Kruskal-Wallis tests that the authors should carefully review in the light of this comment about potential pseureplication.

2. Results interpretation, conclusions and goals

Despite of the fact that the manuscript is generally well-written and structured, I find that the link between results and discussion is still unclear, especially for a general audience with less experience on SDM. In results, the reader can find vast details and analyses, but there are not clear explanations about the implications of these findings. In discussion, the interpretation of the results is often presented in a general manner, hindering the general understanding of the origin of such conclusions. I recommend the authors to make an effort to clarify (in results or discussion; or even better, in both) which specific result in each case enable them to draw a particular conclusion, so that the text gain in clearness and can be accessible for a broader audience. Which specific result helps us to understand that validation is key for SDM? Which one indicates that bioacoustics is “a very effective method for the in situ validation of SDM”? Which one that “we empirically demonstrated that independent field surveys are the best approach to corroborate the predictions made by modelling”?

Moreover, in my opinion, the text would also benefit from a more clear link between the goals presented in the last paragraph of the introduction and the main conclusions presented in the first paragraph of the discussion. The paragraph presenting the study goals lacks an explicit mention to the general aim of the study. The authors did not clearly refer to a key aspect of the study: the assessment of the role of validation methods in SDM and their proposal of using bioacoustic tools as rapid validation method.

3. Methodological aspects that also require clarification

- Historical species records

Authors conducted a literature review for gathering distribution records of the study species. However, some details are missing and prevent the reader from properly understand how this review was carried out. Did authors use keywords in databases as Google Scholar, WoS or Scopus? Did they refine the search by specific field areas? How many documents met the criteria and were reviewed? How many were used to determine the historical records?

- Distribution modelling procedure

This section is well-written, clear and full of details. I particularly appreciate the modelling design applied by the authors that took into account a large number of factors and criteria, and performing a diversity of models. I only have a few minor questions (see the pdf) and a major one: While historical records are taken as presences, how did the authors treat absences and pseudo-absences in the SDM models? I think this should be clarify, considering the significant knowledge shortfalls in species distribution, and the relevant effect of absence in SDM models.

- Acoustic sampling

A large number of details and information should also be added in this section to properly describe some key points (see comments on pdf). Particularly, a key assumption of the study is that the number of sampled days was large enough to get representative data of bat activity in each site. How can we be sure that sampling a minimum of two days enable the authors to determine species presence in a given location?

6. PLOS authors have the option to publish the peer review history of their article (what does this mean?). If published, this will include your full peer review and any attached files.

Reviewer #1: No

Reviewer #2: No

---

## [Author Response · Author response to Decision Letter 0]

13 Aug 2021

Please check the "Response to reviewers" file attached. Thank you.

---

## [Decision Letter · Decision Letter 1]

1 Oct 2021

PONE-D-21-07379R1Bioacoustics for in situ validation of species distribution modelling: An example with bats in BrazilPLOS ONE

Dear Dr. Hintze,

Thank you for submitting your manuscript to PLOS ONE. After careful consideration, we feel that it has merit but does not fully meet PLOS ONE’s publication criteria as it currently stands. Therefore, we invite you to submit a revised version of the manuscript that addresses the points raised during the review process.

We look forward to receiving your revised manuscript.

Kind regards,

Daniel de Paiva Silva, Ph.D.

Academic Editor

PLOS ONE

Journal Requirements:

Additional Editor Comments (if provided):

Dear Hintze et al,

After a new review round, I must say that you are almost there. One of the original reviewers decided for the acceptance of your manuscript, whereas the other decided for a minor review. Therefore, I will grant you all a one-month period in order to proceed with the necessary changes. In case you need more time, please let me know. Still, do not hesitate to submit earlier then the estipulated deadline in case you can.

Best regards,

Daniel Silva, PhD.

Reviewers' comments:

Reviewer's Responses to Questions

**Comments to the Author**

1. If the authors have adequately addressed your comments raised in a previous round of review and you feel that this manuscript is now acceptable for publication, you may indicate that here to bypass the “Comments to the Author” section, enter your conflict of interest statement in the “Confidential to Editor” section, and submit your "Accept" recommendation.

Reviewer #1: All comments have been addressed

Reviewer #3: All comments have been addressed

2. Is the manuscript technically sound, and do the data support the conclusions?

Reviewer #1: Yes

Reviewer #3: Yes

3. Has the statistical analysis been performed appropriately and rigorously? 

Reviewer #1: Yes

Reviewer #3: Yes

4. Have the authors made all data underlying the findings in their manuscript fully available?

Reviewer #1: Yes

Reviewer #3: Yes

5. Is the manuscript presented in an intelligible fashion and written in standard English?

Reviewer #1: Yes

Reviewer #3: Yes

6. Review Comments to the Author

Reviewer #1: I thank the authors for addressing the previous suggestions. This is an interesting contribution combining SDM and bioacoustics, and I congratulate the authors for their work. Finally, I would just suggest adjusting some minor concerns, detailed below.

Figures 5-11: I think that you should reconsider joining all boxplots in a single figure. I suggest that you use accuracy, precision, sensitivity, specificity, g-mean and f-score as lines, and for columns, use LPT, maxSSS, and p10 for repeated for each species. Such as:

Species 1 / Species 2 / Species 3 …

LPT. maxSSS P10 / LPT. maxSSS P10/ LPT. maxSSS P10

Acc

Pre

Sen

Spe

Gme

Fsc

Species 6 / Species 7 / Species 8…

LPT. maxSSS P10 / LPT. maxSSS P10/ LPT. maxSSS P10

Acc

Pre

Sen

Spe

Gme

Fsc

Let the first row with 4 species and the second row with 3.

424: “we proved” is not exactly what you show here, since the performance of the validation is subject to interpretation and depends on many factors, as tuning parameters and thresholds. I believe that the message should be to explore variations in modeling approaches using SDM, since predictions are largely variable even for approaches that are considered optimum.

Reviewer #3: I greatly enjoyed reading this paper, and believe it constitutes an important contribution to the literature on species distribution modeling (SDM), especially in Neotropical region, particularly as it provides important insights on the validation process of SDM.

The article is interesting and overall reads well; however, I have minor comments. The validation approach (based on acoustics) is especially important for non-phyllostomid bats since bats from the Phyllostomidae family have considerable overlap in their call parameters (Yoh et al. 2020). In that sense, I suggest clarifying the main target bat species, which acoustics as validation method, is proper.

Is there any season effect on acoustic surveys explaining absences and potentially affect validation?

Lines 80 – 82. It is true for several families, but it is problematic for Phyllostomidae (Yoh et al. 2020). I suggest clarifying this limitation

Lines 428 -431… It is not true for all bats

Yoh et al, (2020) Echolocation of Central Amazonian ‘whispering’ phyllostomid bats: call design and interspecific variation. Mammal Research

7. PLOS authors have the option to publish the peer review history of their article (what does this mean?). If published, this will include your full peer review and any attached files.

Reviewer #1: No

Reviewer #3: No

---

## [Author Response · Author response to Decision Letter 1]

5 Oct 2021

Please refer to the attached file (Response to Reviewers_Rev2). Thank you.

---

## [Editor Report · Decision Letter 2]

7 Oct 2021

Bioacoustics for in situ validation of species distribution modelling: An example with bats in Brazil

PONE-D-21-07379R2

Dear Dr. Hintze,

We’re pleased to inform you that your manuscript has been judged scientifically suitable for publication and will be formally accepted for publication once it meets all outstanding technical requirements.

Kind regards,

Daniel de Paiva Silva, Ph.D.

Academic Editor

PLOS ONE

Additional Editor Comments (optional):

Dear Hintze et al.

Congratulations! I am pleased to inform you that your manuscript was formally accepted for publication in PLoS One.

Best regards,
---

## [Editor Report · Acceptance letter]

11 Oct 2021

PONE-D-21-07379R2 

Bioacoustics for *in situ* validation of species distribution modelling: An example with bats in Brazil 

Dear Dr. Hintze:

I'm pleased to inform you that your manuscript has been deemed suitable for publication in PLOS ONE. Congratulations! Your manuscript is now with our production department. 

Kind regards, 

on behalf of

Dr. Daniel de Paiva Silva 

Academic Editor

PLOS ONE